# Screen-Printed Electrodes—A Promising Tool for Antineoplastic Drug Detection (Cisplatin and Bleomycin) in Biological Samples

**DOI:** 10.3390/ijms25158030

**Published:** 2024-07-23

**Authors:** Andreea-Cristina Mirica (Ion), Dana Stan, Dragos-Cosmin Zaharia, Horia Iovu, Sorin Mocanu, Marioara Avram, Lorena-Andreea Bocancia-Mateescu

**Affiliations:** 1DDS Diagnostic, 032032 Bucharest, Romania; research.imuno@ddsdiagnostic.com (A.-C.M.); dana_stan@ddsdiagnostic.com (D.S.); zahariadragoscosmin@gmail.com (D.-C.Z.); proteomics@ddsdiagnostic.com (S.M.); 2Advanced Polymer Materials Group, University Politehnica of Bucharest, 1–7 Gh. Polizu Street, 011061 Bucharest, Romania; 3Department of Pneumology I, Carol Davila University of Medicine and Pharmacy, 020021 Bucharest, Romania; 4National Institute for Research and Development in Microtechnologies (IMT Bucharest), 126A Erou Iancu Nicolae Street, 077190 Bucharest, Romania; marioara.avram@imt.ro

**Keywords:** screen-printed electrodes, cisplatin, bleomycin, biosensor, voltammetry

## Abstract

Cancer remains one of the leading causes for death worldwide. Palliative chemotherapy is vital for certain cancer patients, highlighting the critical need for treatment monitoring tools to prevent drug accumulation and mitigate the risk of high toxicity. Therefore, our aim was to evaluate the potential of screen-printed electrodes for the development of sensitive and accurate biosensors for the detection/quantification of antineoplastic drugs. To this purpose, we developed a cisplatin sensor. By functionalizing the gold electrode with human serum albumin and by collecting the electrochemical signal obtained in a H_2_O_2_ solution, through voltammetry measurements, we were able to correlate the current measured at 430 mV with the concentration of cisplatin present in human serum samples, with a correlation coefficient of R^2^ = 0.99. Also, a bleomycin biosensor was developed and proven functional, but further optimization steps were employed in order to improve the accuracy. The developed biosensors have a detection range of 0.0006–43.2 mg/mL for cisplatin and 0.23–7.56 μg/mL for bleomycin in the serum samples. Our preliminary results show that these biosensors can facilitate the real-time monitoring of cisplatin and bleomycin serum levels, allowing healthcare professionals to tailor treatment strategies based on individual patient responses.

## 1. Introduction

Cancer is one of the leading causes of death worldwide and is often treated with highly toxic antineoplastic drugs [1]. Therefore, the treatment monitoring approach is necessary and potentially life-saving, which brings forward the concept of personalized and precision medicine [2]. Due to the clinical importance of therapy monitoring [3], the WHO (World Health Organization) has proposed specific guidelines on how a drug should be evaluated. For 2023, predictions revealed approximately 1,261,990 expected deaths of 702,214 males and 559,776 females in the European Union. Despite these alarming numbers, most cancer types had favorable predicted rates, but, for instance, pancreatic cancer had a stable mortality rate in men but raised 3.4% in women. These predictions led to the conclusion that by increasing the control of certain cancer-inducing factors and by improving screening, early diagnosis and treatments, Europe can achieve a 35% mortality decrease by the year 2035 [4].

Platinum-based drugs, such as cisplatin, are known as antiproliferative agents that were used to treat various types of cancer (ovarian, testicular, head and neck cancer, lung carcinoma, etc.) [5]. The presumed mechanism of action of cisplatin and other members of the platinum-based drugs is thought to be the development of adducts with nucleic acids such as Guanine (G) and Adenine (A), among which 60–65% are intrastrand GG, 25–30% AG adducts, 5% are GNG (where N can be any nucleotide) and only 1–3% interstrand GG [6]. Among the previously described methods for the detection and quantification of platinum-based drugs are high-performance liquid chromatography (HPLC), liquid chromatography–mass spectrometry (LC–MS), inductively coupled plasma mass spectrometry (ICP-MS) and spectrophotometric methods; however, these methods involve the use of expensive equipment and/or the conversion of the drug into a derivate of a similar chemical structure [7].

Bleomycin is a glycopeptide antibiotic that has deoxyribonucleic acid (DNA) cleavage activity upon binding to transition metal ions [8]. The suggested action mechanism involves binding to DNA by the terminal peptide moiety and in the presence of cofactors (metal ions and oxygen) and the induction of single-strand and double-strand damage [9,10]; however, the exact mechanism is not fully elucidated. The methods available for the quantification of bleomycin in biological samples are limited; among them are HPLC with tandem mass spectrometry (HPLC-MS/MS) and ultraviolet–visible spectroscopy (UV-VIS) [11,12,13]. So far, the most sensitive method for detecting as low as 1 ng/mL bleomycin in biological samples is based on ultra-high performance liquid chromatography tandem mass spectrometry (UHPLC–MS/MS), but it requires sample preparation prior to the analysis by solid-phase extraction. However, one recent paper describes an LC-MS-based method that involves a simplified sample processing protocol and good sensitivity and selectivity. The linear range was 15–1500 ng/mL and the correlation coefficient was larger than 0.99 [14]. So far, electrochemical methods have been employed to study the interaction of bleomycin with DNA in the presence of transition metal ions or the interaction between cisplatin and DNA in relation to Guanine and Adenine oxidation signals but not for the development of electrochemical sensors for the quantification of these drugs in biological samples, such as serum and plasma [15,16,17].

The main purpose of our work was to develop screen-printed electrode-based biosensors for the quantification of cisplatin and bleomycin in complex biological samples, such as human serum. To this purpose, we evaluated several functionalization strategies and electrochemical measurements and established two personalized methods for the detection of the antineoplastic drugs. Analytical performances were established by determining the real concentration in the human serum samples and not the concentration in the electrolyte solution, which is much lower.

## 2. Results and Discussion

The electrochemical measurements, including cyclic voltammetry and square wave voltammetry, were employed to assess the functionalization strategies and detection methods for quantifying the two antineoplastic drugs. Additionally, the dispersion of cisplatin on the biosensor surface was analyzed through scanning electron microscopy (SEM).

### 2.1. Electrochemical Detection of Cisplatin in Human Serum

Firstly, we investigated the variations in the current associated with different cisplatin concentrations in the absence of human serum, tested in 5 mM ferric/ferro cyanide potassium solution (Figure 1A) without surface functionalization of the gold electrode. The results show a decrease in the current as the concentration of cisplatin increases and no cisplatin-associated oxidation peak was observed. However, by performing the same experiment, this time on cisplatin-free serum samples (Figure 1B), a more significant decrease in the oxidation and reduction peaks was observed, which suggests that the small modification induced by the cisplatin concentration cannot be detected.

To highlight the oxidation of cisplatin in the absence of serum, we repeated the experiment by changing the electrolyte solution with 10 mM hydrogen peroxide (Figure 2). A recent study reported the development of an electrochemical sensor for tissue monitoring of cisplatin, starting from the premise that cisplatin induces the generation of reactive oxygen species (ROS); cyclic voltammetry (CV) measurements were performed in a H_2_O_2_ solution of a physiological concentration (approx. 10 μM) and a clear correlation between the drug concentration and Ipa (PtII/PtIV) was observed [18].

Because the previous test performed in ferric/ferro cyanide potassium solution had revealed a significant decrease in the oxidation peak, probably caused by the serum components, we decided to block the sensor surface with human serum albumin (HSA), which is also known to bind various drugs, including cisplatin.

Based on the information provided in the literature, shortly after intravenous administration, cisplatin rapidly diffuses into tissues, and more than 90% binds to plasma proteins [19]. This binding affinity of platinum (Pt) drugs to plasmatic proteins has been studied using methods such as high-performance liquid chromatography coupled to inductively coupled plasma-mass spectrometry (HPLC-ICP-MS) and Electrospray Ionization-Quadrupole Time-of-Flight Mass Spectrometry (ESI-qTOF), which made it possible to identify some Pt-containing peptides consisting of cisplatin bound to transferrin (T_f_) and human serum albumin. Furthermore, by comparing the mass spectra of platinated T_f_ and HSA with those of the serum samples, it was obvious that T_f_ and HSA bind to platinum after the incubation of human serum with cisplatin [20].

Based on these insights, we developed the sensor for cisplatin detection by immobilizing human serum albumin on the surface of the gold working electrode and detected the target analyte using the oxidation signal from the drug–H_2_O_2_ interaction.

An increase in the current peak is observed upon addition of 10 μM cisplatin in the 6 mL H_2_O_2_ solution (sample concentration 3.6 mg/mL (1.2 × 10^4^ μM), compared to the peak obtained for the HSA functionalized biosensor, in the absence of the target analyte (Figure 3).

The calibration curve was performed by analyzing 5 μL of serum and the appropriate volume of cisplatin stock solution in order to have in the H_2_O_2_ working solution (6 mL) a final concentration of 0.002 μM, 0.003 μM, 0.1 μM, 0.5 μM, 1 μM, 5 μM, 10 μM, 30 μM, 50 μM, 70 μM, 90 μM, and 120 μM. So basically, these are the detected concentrations in the working solution, but we needed to extrapolate those to the actual serum concentration, thus reaching the concentrations listed in this manuscript: 0.0006 mg/mL (2.4 μM); 0.01 mg/mL (3.6 μM); 0.036 mg/mL (1.2 × 10^2^ μM); 0.18 mg/mL (6.0 × 10^2^ μM); 0.36 mg/mL (1.2 × 10^3^ μM); 1.8 mg/mL (6.0 × 10^3^ μM); 3.6 mg/mL (1.2 × 10^4^ μM); 10.8 mg/mL (3.6 × 10^4^ μM); 18 mg/mL (6.0 × 10^4^ μM); 25.2 mg/mL (8.4 × 10^4^ μM); 32.4 mg/mL (1.0 × 10^5^ μM); and 43.2 mg/mL (1.4 × 10^5^ μM).

The biosensor was obtained by adding 2 μL of a 1 mg/mL HSA solution on the gold working electrode (WE) and incubation for 1h at room temperature. In theory, cisplatin binds to HSA through certain amino acid residues, such as His 105, Met 298, Met 329, etc., and forms an adduct. The detection of cisplatin was carried out by immersing the sensor in a 50 mM H_2_O_2_ solution, which induces an oxidation reaction from Pt(II) to Pt(IV), observed through CV measurements. For a better understanding of the sensor development and testing, Figure 4 shows the scheme, which further explains the sensor functionalization procedure and the working principle of cisplatin detection.

After examining the cyclic voltammetry (CV) plots for each of the assessed concentrations, consistent variations are evident within the 190 mV to 530 mV region (Figure 5). We observed a clear correlation between the measured current at 430 mV and the cisplatin concentrations.

Two sets of calibration curves were prepared using the current versus the cisplatin concentration in human serum. The calibration curves for the low and high concentration range are shown in Figure 6A,B. The equation and R^2^ value of the calibration curves for the low and high concentration range are y = −66.456x^2^ + 40.868x + 4.4178, R² = 0.9936, for low cisplatin concentrations and y = 0.5948x + 12.205, R² = 0.9905, for higher concentrations. Due to the wide quantification range of cisplatin, from 0.6 × 10^−3^ mg/mL to 43.2 mg/mL, two calibration curves were developed to achieve an optimal fit for both the low and high concentrations.

The limit of detection and quantification (LOD and LOQ) were determined using the formulas LOD = 3 σ/m and LOQ = 10 σ/m, where “σ” is the standard error and “m” is the slope of the analytical curve. Thus, we found an LOD of 23 μg/mL and an LOQ of 78 μg/mL for cisplatin in human serum samples. The higher values are attributed to the fact that low concentrations of cisplatin demonstrate poor fitting on a linear curve, instead favoring a second-degree equation for accurate representation. However, the lowest detectable concentration in the electrolyte solution was 30 ng/mL, which can be further improved by optimizing some parameters, such as the volume of the solution in which the target analyte is detected or the concentration of hydrogen peroxide.

For the reproducibility studies, we conducted two experiments, using the same concentrations of cisplatin (0.18 mg/mL–32.4 mg/mL). Accordingly, the RSD values obtained are shown in Table 1.

We utilized linear regression analysis to investigate the relationship between the current and cisplatin concentration in human serum (Table 2). The resulting key parameters are as follows: a slope of 0.73, an intercept of 8.13 and a correlation coefficient of 0.93.

#### 2.1.1. Scanning Electron Microscopy (SEM)

The size distribution of cisplatin in Figure 7 ((A) 0.0006 mg/mL, (B) 10.8 mg/mL, and (C) 43.2 mg/mL) was obtained from the SEM images by measuring around 100 nanoparticles/particles. The SEM images at a magnification of 2000X reveal that the particles exhibit a spherical geometry, and the frequency of the particle distribution and size increases with the concentration of cisplatin. The diameter sizes of the particles for the samples with the concentration of (A) 0.0006 mg/mL, (B) 10.8 mg/mL and (C) 43.2 mg/mL were found to range from 376 nm to 1.41 μm, 351 nm to 2.13 μm and 539 nm to 3.83 μm.

The histogram was best fitted with the Gaussian distribution. The highest percentage of the size of the particles for the sample with the concentration of (A) 0.0006 mg/mL, (B) 10.8 mg/mL and (C) 43.2 mg/mL was found to be within 582–900 nm, 574–1051 nm and 662–1525 nm in the diameter size range, having the mean diameter of 740 ± 184 nm, 819 ± 280 nm and 1097 ± 545 nm.

The SEM images at a magnification of 100kx reveal nanoparticles exhibiting a spherical geometry. The diameter for all the samples ranged from 10 nm to 36 nm, with a mean diameter of 20 ± 5 nm, which is slightly higher for the sample with a concentration of 43.2 mg/mL. These observations align well with the results obtained from the cyclic voltammetry, which indicate an increase in sensor conductivity most likely due to the accumulation of Pt resulting from the formation of HSA-cisplatin adducts. This theory is supported by kinetic studies, revealing that cisplatin and oxaliplatin bind to serum proteins. Distribution studies have shown that the majority of Pt was bound to HSA (96%), while immunoglobulin (IgG) and T_f_ represented minor portions (4%) [21].

#### 2.1.2. Elemental Analysis (EDX)

The distribution of the chemical constituents of interest on the biosensor surface for cisplatin concentrations of 0.0006 mg/mL, 10.8 mg/mL and 43.2 mg/mL was performed with the element energy dispersive spectroscopy (EDS) system (Smart Insight AMETEK). The EDAX map was acquired at an acceleration voltage of 10 kV, with a working distance of 5 mm and 3000× magnification. The EDX spectra revealed signals for nitrogen, oxygen, natrium, platinum and the composition for each cisplatin concentration, as well as the weight and atomic percentage, which are presented in Figure 8, Figure 9 and Figure 10. The peaks of Si and Au are determined by the substrate. The detection of Si in the EDS spectrum suggests that the electron beam passed through the Au electrode and detected the Si given by the substrate, which is a common occurrence in such configurations. The primary challenge arises due to the overlapping M peaks of Au and Pt, which can potentially introduce uncertainty into the quantitative analysis. This overlapping phenomenon complicates the accurate determination of the individual concentrations of Au and Pt in the sample. In addressing this issue, we utilize advanced peak deconvolution algorithms to accurately separate and quantify the contributions of the Au and Pt peaks, enhancing the reliability of the concentration results by resolving the overlapping peaks and assigning them to their respective elements. Additionally, traces of chlorine were also detected in the sample.

### 2.2. Electrochemical Biosensor Development for Bleomycin Detection

#### 2.2.1. DNA Functionalized Sensor

The general opinion is that the antitumor activity of bleomycin is due to the selective cleavage of single- and double-stranded DNA in the presence of oxygen and cofactors, such as metal ions. According to this theory, a specially designed DNA probe was immobilized, which at the 5′ end contains a thiol and at the 3′ end an NH_2_ group. The probe contains a spacer C12, has the sequence 5′–T/Spacer C12/G AGA GAG AGG GGG GGG AAA AAA AAA AAAAAA ACC CCC CCC TCT CTC TCAAAA–3′, has a 53 bp and has a melting temperature of 71.1 °C. A solution of 10 μM DNA probe was prepared, in pure water, free of DNA-ase and RNA-ase, and 2 μL of this solution was pipetted onto the work surface. The sensor was incubated for 2 h at room temperature and then analyzed by CV and electrochemical impedance spectroscopy (EIS) to confirm probe immobilization.

Different concentrations of bleomycin were prepared in phosphate-buffered saline and a 20 mM H_2_O_2_ solution, which were added to the potassium ferric ferrocyanide (5 mM) electrolyte solution. More precisely, in 5 mL of electrolyte solution, 25 μL of H_2_O_2_ and the required volume of bleomycin solution were added to reach the following concentrations: 0.05 mM; 0.06 mM; 0.07 mM; 0.08 mM; 0.09 mM; and 0.1 mM. We conducted the measurements by using EIS (Figure 11), and the results were extrapolated on the calibration curve by associating the concentration with the change in the resistance encountered by an electrode when it is subjected to an external potential or current (Rp). A correlation coefficient R^2^ of over 0.99 was obtained. However, when the developed biosensor was tested with human serum samples, enriched with bleomycin, we obtained a much lower coefficient (R^2^ = 0.79), which suggests significant interference attributed to the serum components.

In order to verify the hypothesis of the cleavage of the DNA macromolecule by the drugs under study (cisplatin and bleomycin), a time evaluation of the interaction between each antineoplastic and DNA templates was attempted by the agarose gel electrophoresis method.

Firstly, the DNA template used was a 356 bp amplicon, obtained using the forward primer 5′–GGT TTG CGG TGT AAG TGC AG–3′ and the reverse primer 5′–AGA CGA GGT CTG CCA TTG TG–3′ that amplify the nsp10 sequence of the SARS-CoV-2 ORF1ab region. Each 20 μL reaction mix consisted of 10 μL master mix (GoTaq^®^ Probe qPCR Master Mix with dUTP, Promega), 0.4 μL of Taq Polymerase (GoScript™ RT Mix for 1-Step RT-qPCR, Promega), 1 μL from each primer (stock solution 10 μM), 2.6 μL of nuclease-free water and 5 μL of template, and the PCR amplification program was 48 °C for 15 min, 94 °C for 3 min, 95 °C for 15 s and 60 °C (40×). DNA template purification was carried out using the Wizard^®^ PCR Preps DNA Purification system, provided by Promega, Madison, USA. Then, six DNA template samples were prepared, two supplemented with 10 mM cisplatin, two with 10 mM bleomycin and two control tubes with no drug.

The samples were loaded in 1.5% agarose gel, using loading dye Orange G and stained with ethidium bromide. The gel size was 10x10 cm and the running time was ~30 min at 50 V. The Invitrogen 50 bp DNA Ladder, provided by ThermoFisher scientific, in the range of 50 bp to 2500 bp was used to assess the amplicon size and potential changes due to the drug interaction. Initially, the evaluation was established for an interval from 2 to 24 h, but the results showed that there were no significant changes in the quality of the DNA, nor in the size of the bands obtained, as a result of a possible link between the drugs and the DNA template. In the next stage, the samples were incubated for a longer period of time, from 24 h to 72 h (Figure 12). According to this evaluation, even after 72 h of incubation, no significant changes can be observed in the amplicon size.

#### 2.2.2. Bleomycin Resistance Protein Functionalized Sensor

According to the literature studies, the binding mechanism of the bleomycin resistance protein to bleomycin (BLE) is not fully elucidated. There are several hypotheses that the interaction of the bleomycin resistance protein (BRP) with BLE is analogous to the interaction of BLE with DNA [22]. The bleomycin resistance protein has been proven effective in binding molecules similar to bleomycin, thereby inhibiting bleomycin-induced DNA degradation. In silico modeling has confirmed that the mechanism of action involves the dimerization of the BRP to sequester bleomycin and prevent DNA damage. The BRP exhibits specificity toward bleomycin-like molecules, as the cloning and expression of the bleomycin resistance gene, ble_MBL_, in *Staphylococcus aureus* did not impart cross-resistance to other antimicrobial glycopeptides, such as vancomycin and teicoplanin [23].

The electrochemical behavior of bleomycin at the gold screen-printed electrode functionalized with the bleomycin resistance protein in the supporting electrolyte solution (5 mM solution of K_4_[Fe(CN)_6_]/K_3_[Fe(CN)_6_]) was examined using SWV measurements (Figure 13A). The measured current (I) appears to decrease after the binding of bleomycin to the functionalized BRP electrode from 5 μA (BRP) to 4.2 μA (BLE in PBS).

Similar behavior is exhibited when the gold electrode, modified with the bleomycin resistance protein, is exposed to human serum (Figure 13B). This resemblance may be known by considering that the gold surface is not fully saturated with BRP, allowing for proteins and other serum molecules to bind to the unoccupied regions of the electrode surface. Contrary to the experiments performed with BLE-free serum or BLE, the binding of BLE in serum with BRP determines an increase in the current, potentially attributed to interactions between the serum and bleomycin (Figure 13C).

The electrochemical interaction between human serum and bleomycin is not extensively studied in the literature, and the specific details of such interactions may be limited. Nevertheless, the existence of particular ions or biomolecules within serum could initiate redox reactions that can lead to changes in the chemical properties of bleomycin. The interaction of bleomycin with metal ions is a critical aspect of its mechanism of action. Bleomycin contains a metal-binding domain, typically coordinated to a metal ion, such as iron (Fe) or copper (Cu) [24]. According to the manufacturer (BioRad) of the human serum control we used in the evaluation of the bleomycin sensor, the level of Cu is about 62.3 μg/dL and for Fe it is 63 μg/dL. Bleomycin binds to metal ions and this complex can activate molecular oxygen leading to the cleavage of DNA strands, which is a key aspect of bleomycin’s cytotoxic activity against cancer cells [15].

The voltammograms represented in Figure 14 illustrate the quantification of BLE in serum across a range of concentrations from 0.23 μg/mL to 7.56 μg/mL. The differential current increases with the BLE concentrations from 3.93 μA (0.23 μg/mL) to 6.8 μA (7.56 μg/mL), which confirms that bleomycin can be quantitatively detected in human serum by the proposed protocol.

The calibration curve is illustrated in Figure 15. We obtained a correlation coefficient (R^2^) of 0.99 and an LOD and LOQ of 1.1 μg/mL, respectively, 1.25 μg/mL, which makes this method suitable for bleomycin detection and quantification in human serum samples; however, the sensitivity can be improved by further testing and calibration schemes.

The reproducibility analysis of the bleomycin biosensor involved measurements of the following concentrations of BLE, 1.52 μg/mL, 7.56 μg/mL and 15.12 μg/mL, utilizing two distinct sensors (Table 3). The RSD values are between 2.9% and 11.6%.

### 2.3. Comparison of the Current vs. Previously Developed Biosensors for Cisplatin and Bleomycin

The analytical performances of the developed biosensors for cisplatin and bleomycin detection were compared with some of the data presented by other authors and the results are summarized in Table 4. While some of the sensors developed so far have an impressive LOD and were proven to detect the target analyte with great accuracy in a variety of buffers, they also have some shortcomings such as a narrow linear range or lack the evaluation with various target analyte concentrations in biological samples in order to demonstrate their accuracy and sensitivity in the presence of complex multi-interference matrices. The LOD obtained for the HSA/GE sensor for cisplatin detection is higher than the one obtained by Petrlova et al., 2006, but the serum sample was diluted 1000 times, while we used undiluted serum samples, and they have a lower detection range [25]. The LOD for the BRP/GE biosensor was similar to the SH-DNA1/GE sensor; however, we managed to obtain an extended linear range. Although the sensors developed in the current study do not always exceed the analytical performance of those reported in the specialized literature, they were designed and calibrated by taking into account the real needs of the clinical laboratories, the calibration curves were made on human serum and the tested concentrations extrapolated correctly for the biological samples.

## 3. Materials and Methods

### 3.1. Reagents and Solutions

Chemical reagents, including potassium hexacyanoferrate (II) trihydrate (98.5%) and potassium hexacyanoferrate (III) (99%), were procured from VWR Chemicals, Radnor, PA, USA. Lyophilized powder of human serum albumin (96%) was acquired from Alfa Aesar, Ward Hill, MA, USA. Cisplatin, bleomycin sulfate, phosphate-buffered saline (PBS), hydrogen peroxide (H₂O₂, 30%) and nuclease-free water for molecular biology were obtained from Sigma Aldrich, St. Louis, MI, USA. The DNA probe with the sequence 5′–T/Spacer C12/G AGA GAG AGG GGG GGG AAA AAA AAA AAAAAA ACC CCC CCC TCT CTC TCAAAA–3′ and primers were procured from Biolegio, Nijmegen, Netherlands. The GoTaq^®^ Probe 1-Step RT-qPCR System and Wizard^®^ PCR Preps DNA Purification system were provided from Promega Corporation, Madison, WI, USA. The SARS-CoV-2 culture fluid heat-inactivated isolate USA-WA1/2020 was from ZeptoMetrix Corp. Buffalo, NY, USA. The agarose was purchased from Melford, Ipswich, UK, and the Invitrogen 50 bp DNA Ladder was procured from ThermoFisher scientific, Waltham, MA, USA. The bleomycin resistance protein, NCBI accession NP_040432.1, was provided from MyBiosource (MBS1449950) Southern California, San Diego, CA (USA), and the human serum-based control (Lyphochek Assayed Chemistry Control) was supplied by Bio-Rad Laboratories, Hercules, CA (USA).

### 3.2. Apparatus

For this study, we used screen-printed electrodes (SPEs) with a gold (Au) ink working electrode manufactured by NANOM MEMS SRL, Râșnov, România. The gold electrodes were printed with a semiautomatic screen-printer (LC-TA-250 Model) using alumina 96% as the substrate (0.6 mm thickness). The working electrode and the counter electrode were made of Au ink, and the pseudo-reference electrode was made of silver (Ag) ink. The Au and Ag inks were printed and heat-treated at 850 °C for 15 min. The conductive paths connecting the working electrode, the counter electrode and the pseudo-reference electrode were protected with a dielectric layer obtained from a borosilicate glass paste. All the pastes were made by NANOM MEMS. The gold screen-printed electrodes used for this study had the following technical specifications: thickness: 1 mm; length: 5 cm; width: 1 cm; substrate: alumina; contacts and paths: silver paste; coating: chemical resistant dielectric paste; working electrode: area 12 mm^2^, paste: gold; auxiliary or counter electrode: area 30 mm^2^, paste: gold; and reference electrode: area 0.6 mm^2^, paste: Ag.

The electrochemical experiments were performed using a potentiostat from GAMRY Instruments. The interface 1010E is fully functional and can perform all the techniques of voltammetry measurement and electrochemical impedance spectroscopy up to 2 MHz with temperature monitoring. For the cyclic voltammetry (CV), electrochemical impedance spectroscopy (EIS) and square wave voltammetry (SWV) measurements, the software Echem Analyst version 7.07 was used. The obtained data were analyzed and processed using OriginPro v. 8.5.

The scanning electron microscopy (SEM) images of the cisplatin biosensor, after the addition of different analyte concentrations, were acquired with a Nova NanoSEM 630 Scanning Electron Microscope (FEI Company, Hillsboro, OR, USA), using a UHR detector (Through-Lens-Detector-TLD) at an acceleration voltage of 10 kV. To determine the elemental composition of the surfaces, TEAM™ EDS Smart Quant and Element version V4.4.1 software was employed to identify the elemental composition of the surfaces before and after evaluating the serum samples with different concentrations of cisplatin.

### 3.3. Functionalization of the SPE for the Cisplatin Detection

The initial characterization of the electrodes was carried out using cyclic voltammetry, with a scanning rate of 100 mV/s and the potential range between −0.200 V and 0.600 V, impedance spectroscopy and square wave voltammetry techniques in a solution of ferric/ferro cyanide potassium K_4_[Fe(CN)_6_]/K_3_[Fe(CN)_6_] 5 mM. After the initial characterization, the electrode was washed with PBS solution pH 7.4 and left to completely dry at room temperature. Cleaning of the electrode was performed as described by a previous paper, using H_2_O_2_ solution and the CV technique [30].

The functionalization of the working electrode surface was performed by adding 2 μL of human serum albumin (HSA) from a 1 mg/mL solution in PBS buffer (pH 7.4), followed by 1 h incubation, at room temperature. After incubation, the electrode was washed four times with PBS solution, pH 7.4, and characterized both in ferric/ferro cyanide potassium 5 mM and H_2_O_2_ 50 mM solution, using CV, with a scan rate of 250 mV/s and a potential range of −0.200 V and 0.800 V. In order to confirm the immobilization of the protein on the gold surface, EIS and SWV were also performed. Following these analyses, the electrode underwent another four washes with PBS solution and was left to dry.

The cisplatin concentrations were prepared in the range of 0.0006 mg/mL and 43.2 mg/mL and analyzed in a fresh H_2_O_2_ 50 mM in PBS pH 7.4 solution; the measurements were conducted using CV.

### 3.4. Functionalization of the SPE for the Bleomycin Detection

The initial characterization of the screen-printed electrode (SPE) followed the same protocol as that used for the biosensor designed for cisplatin detection.

The functionalization of the gold electrode was performed by adding 2 μL of bleomycin resistance protein (BRP) solution of 0.6 mg/mL on the surface of the working electrode and incubated at room temperature for 1 h. Subsequently, the excess was washed with pH 7.4 PBS buffer, and the efficacy and electrode resistance of the modified sensor were evaluated. The electrochemical measurements of the electrode surface were performed in 5 mM solution of K_4_[Fe(CN)_6_]/K_3_[Fe(CN)_6_] by CV in the potential range of −0.200 V and 0.600 V, with a scanning rate of 100 mV; EIS measurements using a frequency range of 0.1 mHz to 100 kHz, an amplitude of 10 mV and a potential of 10 mV (AC Voltage mV rms); and SWV in the interval −0.05 V and 0.4 V. Various concentrations of bleomycin in human serum ranging from 0.23 μg/mL to 7.56 μg/mL were prepared, and the binding capacity of bleomycin to the biosensor was evaluated by SWV measurements in the range −0.05 V and 0.4 V. SWV was employed for the quantitative determination of bleomycin due to its increased sensitivity compared to CV and EIS.

## 4. Conclusions

Our preliminary results revealed that the cisplatin sensor can accurately detect a wide range of concentrations (0.0006 mg/mL to 43.2 mg/mL) in human serum, with a detection limit of 23 μg/mL and minimum interference from the matrix components, while the bleomycin sensor, although not as efficient, can detect between 0.23 and 7.56 μg/mL. Compared to other recently developed electrochemical biosensors for antineoplastic drug detection, the ones developed by us may not always be the most performant; however, considering the complex matrix, that is, the human serum, these results are very promising. Future studies will aim at increasing the sensitivity by reducing the interference and optimizing the working procedure.

In our study, regarding the drugs’ ability to cleave DNA, there was no obvious modification after 24–48 h; only after 72 h did the size of the tested amplicons seem to vary a bit, but the degree of degradation could not be clearly established. This could be explained by the fact that the main action mechanism of cisplatin is not exhibited by DNA cleavage but by binding to the macromolecule and disrupting the replication and transcription processes, which can lead to cell death caused by cell responses to cisplatin-induced damage.

In summary, the development of electrochemical biosensors for antitumor agents addresses the critical need for personalized monitoring in oncology, aiming to enhance treatment efficacy and patient outcomes through the precise control of drug levels in the bloodstream. Alternatively, they could be used to assess the residual drug concentration in the body, after elimination through the urinary tract, by comparing the amount of drug in the patient urine to the administered drug dose.

## Figures and Tables

**Figure 1 ijms-25-08030-f001:**
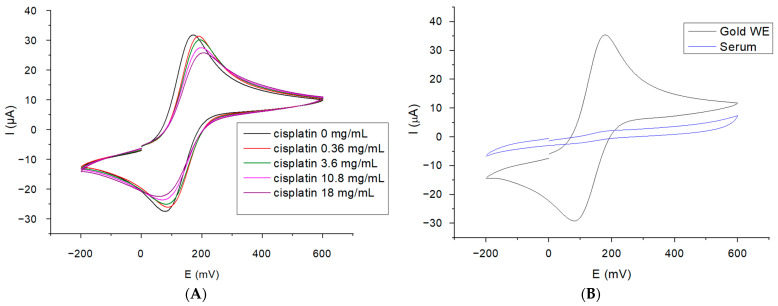
(**A**) The cyclic voltammograms obtained for the cisplatin solutions in PBS ranging from 0 to 18 mg/mL, analyzed in 5 mM solution of K_4_[Fe(CN)_6_]/K_3_[Fe(CN)_6_]; (**B**) the cyclic voltammograms for the human serum binding to the gold working electrode surface in 5 mM solution of K_4_[Fe(CN)_6_]/K_3_[Fe(CN)_6_].

**Figure 2 ijms-25-08030-f002:**
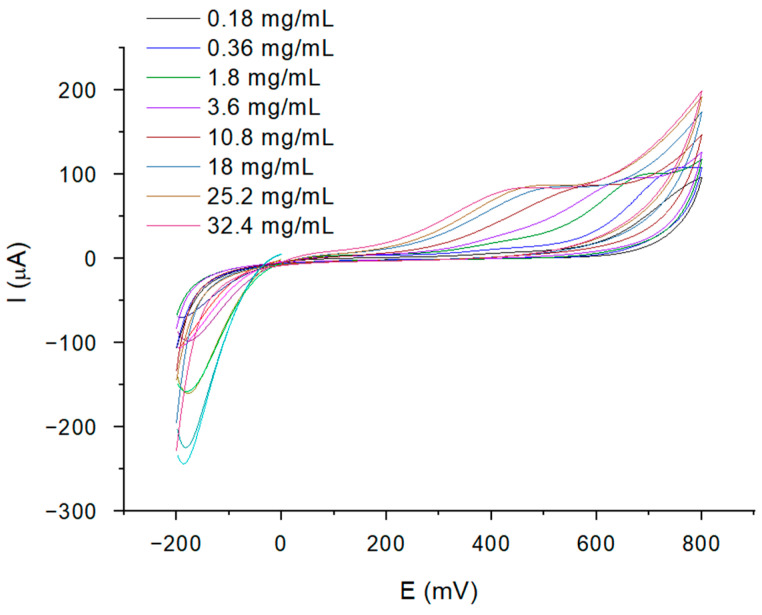
CV plots for the cisplatin concentrations between 0.18 mg/mL and 32.4 mg/mL (in PBS) without HSA modification of the electrode, characterized in H_2_O_2_ 10 mM.

**Figure 3 ijms-25-08030-f003:**
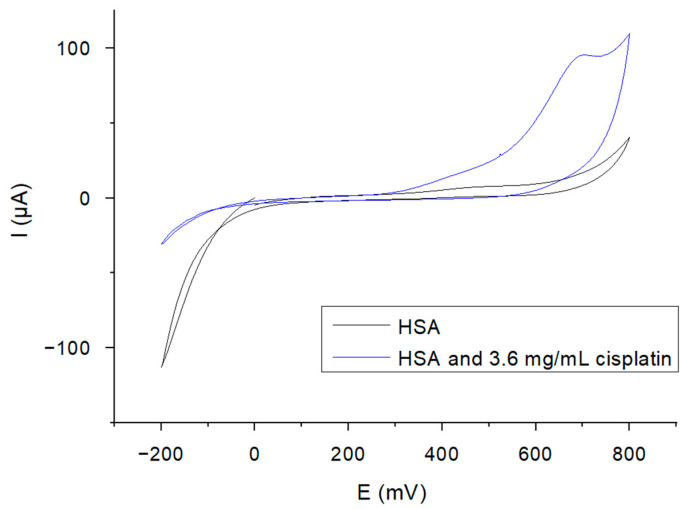
CV plots of gold electrodes functionalized with HSA and 3.6 mg/mL cisplatin detection in 50 mM solution of H_2_O_2_.

**Figure 4 ijms-25-08030-f004:**
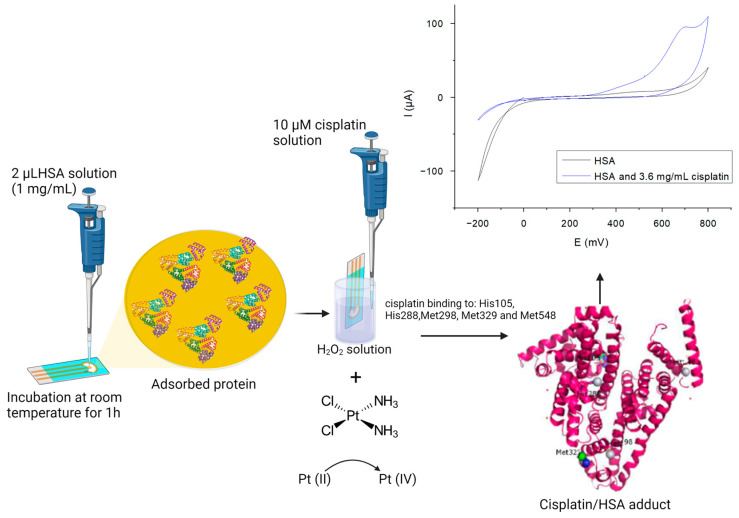
Cisplatin biosensor development.

**Figure 5 ijms-25-08030-f005:**
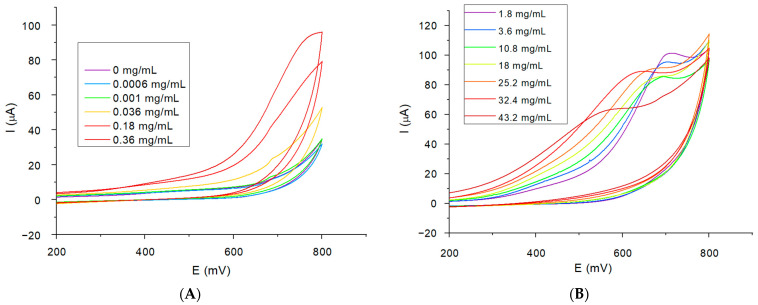
(**A**): Cyclic voltammograms of cisplatin concentrations ranging from 0 mg/mL (HSA functionalized electrode) to 0.36 mg/mL in 50 mM solution of H_2_O_2_ and (**B**): cyclic voltammograms of cisplatin concentrations between 1.8 mg/mL and 43.2 mg/mL in 50 mM solution of H_2_O_2_.

**Figure 6 ijms-25-08030-f006:**
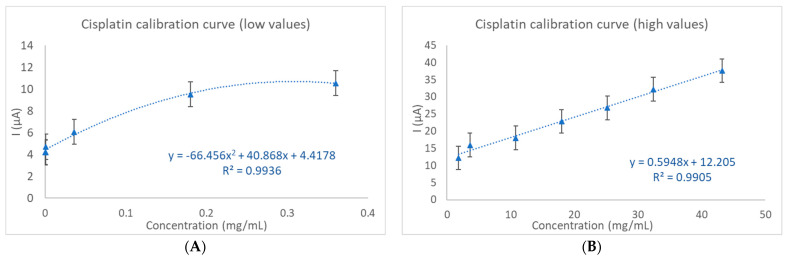
The calibration curve for the cisplatin detection and quantification sensor for (**A**): low values and (**B**): high values.

**Figure 7 ijms-25-08030-f007:**
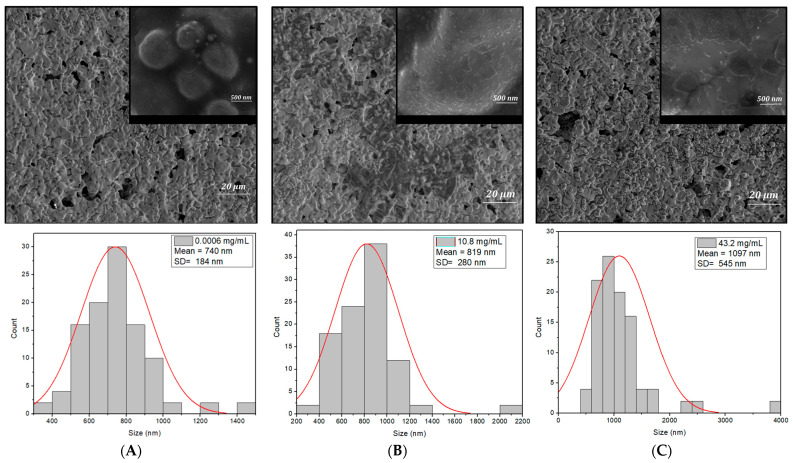
The SEM micrographs and distribution of the biosensor after the addition of the cisplatin concentrations: (**A**) 0.0006 mg/mL, (**B**) 10.8 mg/mL and (**C**) 43.2 mg/mL.

**Figure 8 ijms-25-08030-f008:**
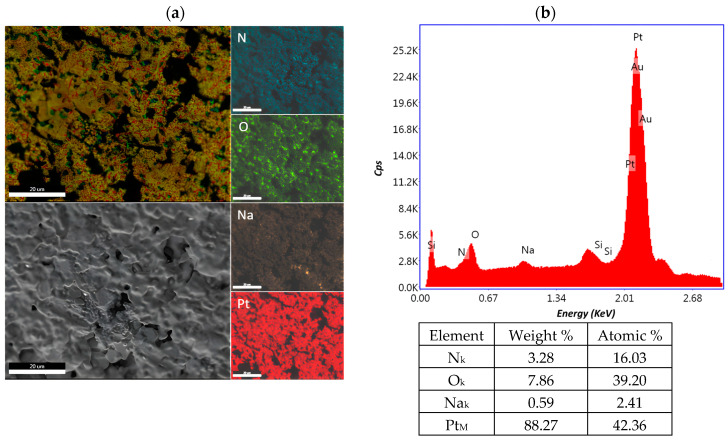
Element distribution mapping at 3000× magnification of the 0.0006 mg/mL sample. (**a**) An SEM image of the investigated area and EDX mapping of the pattern of the element distribution and (**b**) quantitative analyses spectrum.

**Figure 9 ijms-25-08030-f009:**
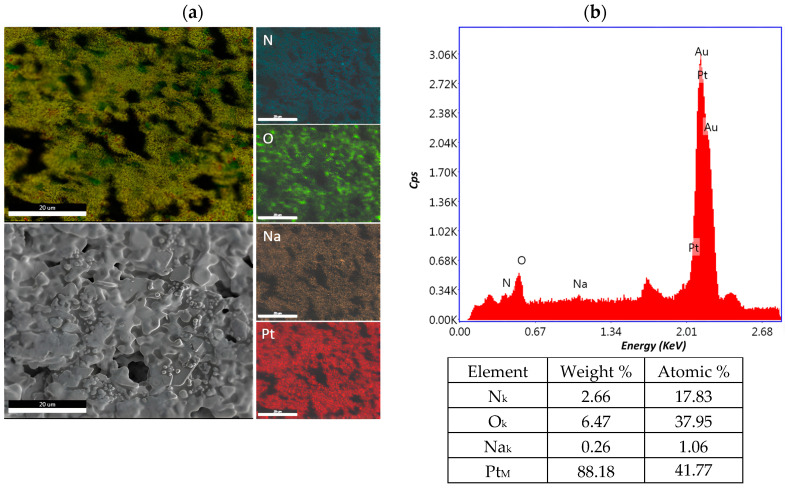
Element distribution mapping at 3000× magnification of the 10.8 mg/mL sample. (**a**) An SEM image of the investigated area and EDX mapping of the pattern of the element distribution and (**b**) quantitative analyses spectrum.

**Figure 10 ijms-25-08030-f010:**
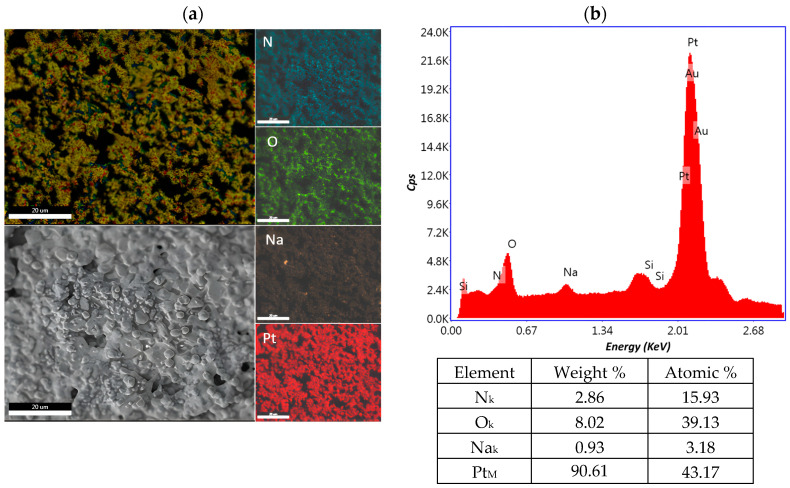
Element distribution mapping at 3000× magnification of the 43.2 mg/mL sample. (**a**) An SEM image of the investigated area and EDX mapping of the pattern of the element distribution and (**b**) quantitative analyses spectrum.

**Figure 11 ijms-25-08030-f011:**
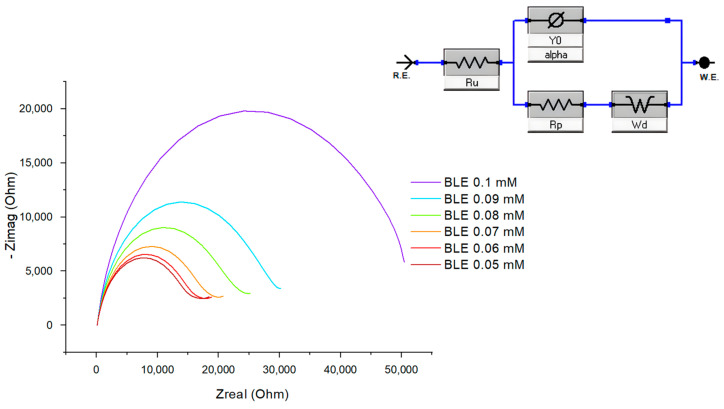
Nyquist diagrams of bleomycin concentrations ranging from 0.05 mM to 0.1 mM in potassium ferric ferrocyanide/H_2_O_2_ solution.

**Figure 12 ijms-25-08030-f012:**
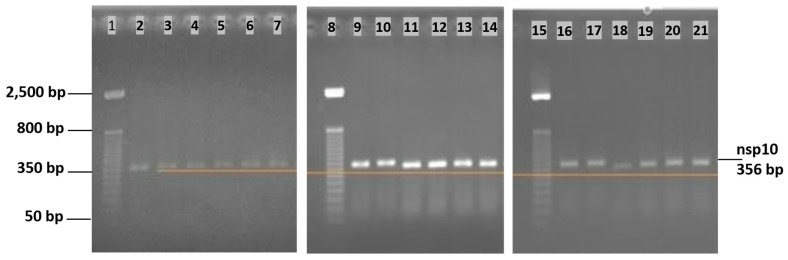
Agarose gel electrophoresis: Lane 1—DNA ladder; Lanes 2–3—cisplatin samples after 24 h; Lanes 4–5—bleomycin samples after 24 h; Lanes 6–7—blank samples; Lane 8—DNA ladder; Lanes 9–10—cisplatin samples after 48 h; Lanes 11–12—bleomycin samples after 48 h; Lanes 13–14—blank samples; Lane 15—DNA ladder; Lanes 16–17—cisplatin samples after 72 h; Lanes 18–19—bleomycin samples after 72 h; and Lanes 20–21—blank samples.

**Figure 13 ijms-25-08030-f013:**
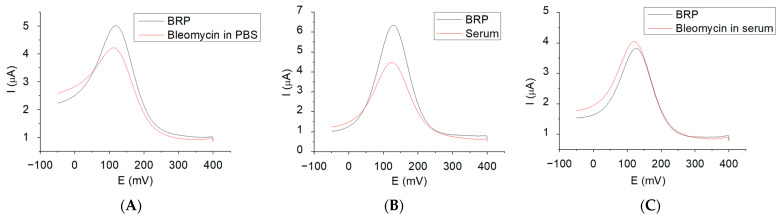
The SWV plots of (**A**) BRP and 1.51 μg/mL BLE in PBS; (**B**) BRP and human serum on the gold working electrode; and (**C**) BRP and bleomycin in serum in 5 mM solution of K_4_[Fe(CN)_6_]/K_3_[Fe(CN)_6_].

**Figure 14 ijms-25-08030-f014:**
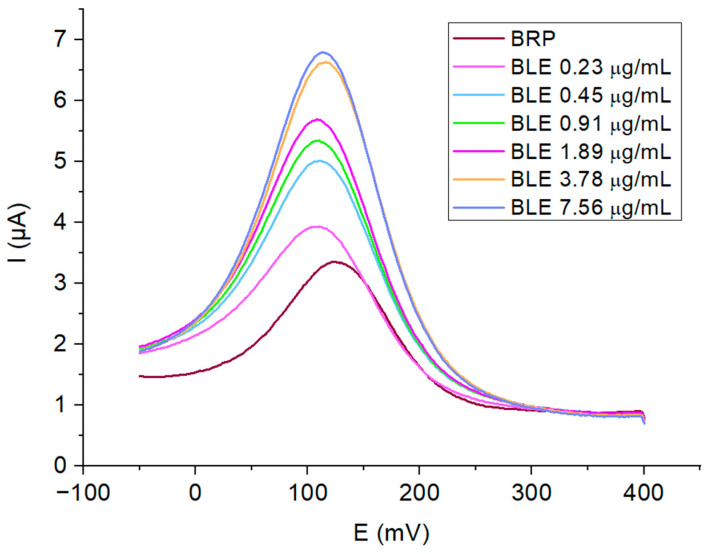
The SWV plots of the bleomycin concentrations ranging from 0.23 μg/mL to 7.56 μg/mL.

**Figure 15 ijms-25-08030-f015:**
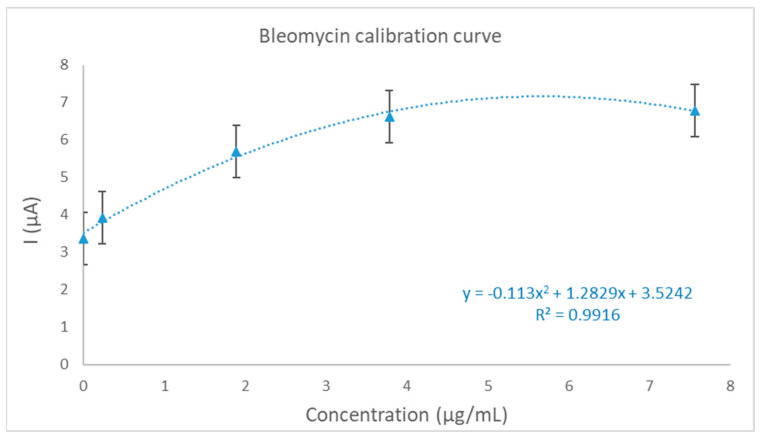
The calibration curve for BLE quantification in the serum samples.

**Table 1 ijms-25-08030-t001:** The reproducibility study for the cisplatin electrochemical biosensor.

Conc. (mg/mL)	Experiment 1 (Different Sensors)			Experiment 2 (Same Sensor)			Experiment 3 (Different Sensors)			Average	STDV	RSD	RSD%
	I_conc._ (μA)	I_blank_ (μA)	I_conc._–I_blank_	I_conc._ (μA)	I_blank_ (μA)	I_conc._–I_blank_	I_conc._ (μA)	I_blank_ (μA)	I_conc._–I_blank_				
0.18	9.52	8.1	1.42	4.8	3.57	1.23	6.87	4.99	1.88	1.51	0.33	0.22	22.13
3.6	15.94	6.35	9.59	13.83	3.57	10.26	20.33	8.51	11.82	10.56	1.14	0.11	10.84
10.8	18.05	4.42	13.63	18.29	3.57	14.72	21.52	8.80	12.72	13.69	1.00	0.07	7.31
18	22.87	7.31	15.56	25.17	3.57	21.6	27.42	1.98	25.44	20.87	4.98	0.24	23.87
32.4	32.21	5.81	26.4	40.44	3.57	36.87	39.17	1.98	37.19	33.49	6.14	0.18	18.33

**Table 2 ijms-25-08030-t002:** Regression statistics of the cisplatin calibration curve.

Regression Statistics	
Multiple R	0.966
R square	0.934
Adjusted R square	0.928
Standard error	2.992
Intercept (mg/mL)	8.133
Slope ($\frac{\Delta \left(\mathrm{mg}/\mathrm{mL}\right)}{\Delta \left(\mu A\right)\;}$)	0.732
Standard deviation (intercept)	1.032
Standard deviation (slope)	0.059

**Table 3 ijms-25-08030-t003:** The reproducibility study for the bleomycin electrochemical biosensor.

Conc. (μg/mL)	Experiment 1	Experiment 2	Experiment 3	Average	STDV	RSD	RSD %
	I (μA)	I (μA)	I (μA)				
1.52	4.22	4.05	3.93	4.07	0.15	0.04	3.58
7.56	5.76	6.79	5.90	6.15	0.56	0.09	9.08
15.12	6.78	7.68	9.18	7.88	1.21	0.15	15.39

**Table 4 ijms-25-08030-t004:** A comparison of the analytical performances of the cisplatin and bleomycin electrochemical biosensors.

Electrode	Detected Compound	Linear Range (μmol L^−1^)	LOD (μmol L^−1^)	Matrix	Ref.
MB-P-oligo-AG DNA/GE	Cisplatin	0.2–2.0	0.20	pH 5 HEPES buffer	[6]
PtMT-HMDE	Cisplatin	10–160 350–650	data not found	human blood serum diluted 1000 times	[25]
CdTc-HMDE	Cisplatin	10–650	2.50	human blood serum diluted 1000 times	[25]
MWCNT-COOH/SDS/SPE	Cisplatin	1.45 × 10^−5^–1.0 × 10^−4^	4.60	0.1 mol L^−1^ NaCl	[26]
GO-MWNTs/GCE	Cisplatin	1.30–26.0	0.11	5% human serum	[27]
HSA/GE	Cisplatin	2.0–14.4 × 10^4^	76.00	human serum	This work
SH-DNA1/GE	Bleomycin	0–1.0	0.1	D-buffer	[28]
pSPCE/PbNPs	Bleomycin	0.0001–0.02	2.8 × 10^−5^	NaAc—HAc solution (pH 4.5)	[29]
BRP/GE	Bleomycin	0.1–5.0	0.7	human serum	This work

## Data Availability

The original contributions presented in the study are included in the article, further inquiries can be directed to the corresponding author.

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
