# Peer review of "Screen-Printed Electrodes—A Promising Tool for Antineoplastic Drug Detection (Cisplatin and Bleomycin) in Biological Samples"

_ijms, 2024, doi:10.3390/ijms25158030_

Round 1

Reviewer 1 Report

Comments and Suggestions for Authors

In this work, the authors r evaluated the potential of screen-printed electrodes for the development of sensitive and accurate biosensors for the detection/quantification of antineoplastic drugs. The results show that these biosensors can facilitate real-time monitoring of cisplatin and bleomycin serum levels, allowing healthcare professionals to tailor treatment strategies based on individual patient responses. The work is interesting but needs a bit revision before it will be suitable for publication.

1.       The font is not uniform in table 2. Please unify the font.

2.       R2 in the SWV results in Figure 14 is too low.

3.       Please supplement error bar analysis.

4.       The CV Curve is not closed, so it is recommended to optimize it.

5.       The image quality is too low. It is recommended to redo it.

6.       EDX is not as accurate as XPS. Please supplement XPS results.

Author Response

Esteemed reviewer,

On behalf of the authors, I want to thank you for taking the time to evaluate our work and for all your valuable suggestions and comments. Attached, you can find a document with detailed description of the modifications.  

Kind regards,

PhD. Lorena Bocancia-Mateescu

Reviewer 2 Report

Comments and Suggestions for Authors

Current research developed a cisplatin sensor and a bleomycin sensor to allow real-time monitoring of cisplatin and bleomycin serum levels, allowing healthcare professionals to tailor treatment strategies based on individual patient responses. Still, there are certain kinds of issues that the authors need to address or to justify before publication.

1. Page 3-5, Figure 1-3,5. The in-figure legends and labels for each axis are blurry. Also, why are there two color patterns corresponding to the same sample?

2. Page 4, Figure 4, legends and labels are difficult to recognize from the sub-graphs.

3. Page 5, Figure 7, the upper panel lacks scale labeling and lower panels are blurry. 

4. Page 8-10, Figure 8-10, scale labeling is blurry for panel a and panel b should be replaced with a clearer presentation.

5. Page 10, Figure 11, the in-figure text can not be read from “Equivalent circuit scheme”

6. Page 11-12, Figure 12-13, labeling in-figure is difficult to read. 

Author Response

Esteemed reviewer,

Thank you for taking the time to evaluate our work. We appreciate all your valuable suggestions and comments. I have attached a document which contains detailed responses to all the comments. 

Kind regards,

The authors 

Reviewer 3 Report

Comments and Suggestions for Authors

The present manuscript shows the suitability of a cyclic voltametry method for the use of electrodes treated with serum albumin and the use of a redox reaction for the monitoring of the anticancer metallo drugs cysplatin and bleomycin.  The utility of this method might be the simplicity, low sample amounts.  The study is very preliminary, the limits of detection are higher than what would be required in a clinical setting, the selectivity of the current curves will still need to be established.  Also it is not clear to me that the method can only detect free cisplatin, as opposed to total cisplatin.

In lines 38-41 the sentence makes no sense (not important).  Pancreatic cancer is not the sole type of cancer with bad outcomes of survival (few months or years).  Cancers of the stomach, brain, some lymphomas, breast and testicular can all be fatal.

Figures 1 thorugh 5 and Fig 13 have poor resolution, so they need to be improved..  I had to zoomed them by 300% to make the details clear.  Figure 6 in contrast has good resolution in the original size.  For Figure 1 there are two experiments described in Figure1B but 4 different color lines, without explanation.  The same follows for the other CV graphs, what is the meaning of the unannotated curves?

The scheme shown in Figure 4 shows the basis of the method for the CV-based detection of cisplatin, yet it needs to be explained in the text.

Table 1 shows the reproducibility of the measurements. However the number of repeats needs to be higher (3-5).  Duplicate sets of measurements are not sufficient for biological samples given the variability in the samples.  The same follows for Table 3

The parameters described in Table 2 have not units for the slope and intercept.  They need to be indicated there.

The SEM images in Figure 7 do not have the scale bars that are common in this type of microscopy measurements.  They need to be added.

In line 202 the authors suggest cisplatin reacts much faster with serum proteins. But this is compared to what?  HSA?  The sentence needs clarification.

EIS needs to be described when introduced in line 243.  HSA needs to be described before Figure 1.  BRP is introduced but not defined in page 11 in line 293.

Is there a reason why there is a Nyquist diagram shown for bleomycin and not for cysplatin.  What is the meaning of the engineering treatment in this context?

Figure 13 is not explained in sufficient detail.  The SWV plots do not show a particular effect of BLE.  The conditions for the voltamograms are not explained in detail.  The authors invoke the chemistry of Fe-BLE as responsible for the observed activity (63 ug/dL).  But the Fe concentration is total, not chelatable by BLE.  Did the authors correct for this fact or is it not important, and why?  Is the meaning of Figure 13B that serum proteins attenuate the BRP binding to the electrode?

In Line 354, the reference number No 25 is not indicated.

Figure 14, which also needs its resolution increased, shows the BLE dependence of the wave amplitude.  How do we know this response is specific for BLE in actual (not commercial serum) samples?

There are quite a few text corrections and clarifications that need to be made and which I indicate below:

Line:                        Text says:                           Change to:

47                           believed to be                   though to be

57                           after binding                     upon binding

68                           analytical domain              linear range

68                    coefficient of determination    correlation coefficient

80                           obviously                           (delete)

95                    could not be detected             can't be detected

111                        decreased                          decrease

119                       ESI-Q-TO                             ESI-qTOF

125                         sur-face                             surface

there a lots of words truncated by the processor or during translation.  Please check the entire manuscript

144         made a graphical representation     figure 4 shows the scheme

148         graphical representation of              (delete)

150         190 -530 mV                                    190 to -530 mV

166        standard error                                   standard deviation of the blanks

175        are in                                                  are shown in

180        coefficient of determination            correlation coefficient

185        2kX                                                   2000X

192       Gauss function                                 Gaussian distribution

249       EIS                                      Electrochemical Impedance Spectroscopy

292       elucidated                                         known

363       Matrice                                             Matrix

Finally, I am confused as to why the cisplatin electrochemical method of detection used a potential of 430 mV which does not show maximal current changes in Figure 5.

Comments on the Quality of English Language

The English is adequate but some term usage needs to be corrected as shown in the review.

Author Response

Esteemed reviewer,

I want to thank you on behalf of all authors for taking the time to evaluate our work. We found your suggestions and comments very helpful and believe that they had a positive impact on the manuscript. Please find attached a detailed response to all the comments.

Kind regards,

PhD. Lorena Bocancia-Mateescu

Round 2

Reviewer 1 Report

Comments and Suggestions for Authors

The author has revised the article as required, so we believe that the article meets the requirements for publication in the journal.